# Mechanical Modelling of the Strength and Stiffness of Circular Hollow Section Tube under Localised Transverse Compression and Tension

**DOI:** 10.3390/ma16072641

**Published:** 2023-03-27

**Authors:** Massimo Latour, Sabatino Di Benedetto, Antonella Bianca Francavilla, Alberico Saldutti, Gianvittorio Rizzano

**Affiliations:** Department of Civil Engineering, University of Salerno, 84084 Fisciano, Italy

**Keywords:** Circular Hollow Sections, through-all plates, strength, stiffness, FEM, parametric analysis, regression analysis, component method, 3D Laser Cutting Technology

## Abstract

The component method is a powerful tool for designing and modelling steel beam-to-column connections. Its widespread use is ensured by several formulations currently included in Eurocode 3 part 1.8 for welded and bolted joints. However, the recent use of 3D Laser Cutting Technology (3D-LCT) in the construction market has enlarged the range of solutions, allowing the realisation of tubular columns with passing-through elements. Given the recent development, no design formulations are currently provided for this typology. At this moment, only a few research studies have developed to fill this knowledge gap. At the University of Salerno, since some years, research efforts are ongoing to characterise the flexural strength of connections between Circular Hollow Section columns and passing double-tee beams, suggesting methodologies to predict the behaviour of the resistance and stiffness of this typology and some of its elementary joint components. Within this framework, this paper aims to examine the strength and stiffness of one of the main components of this joint, which was never examined previously, that is the so-called tube under localised transverse tension/compression. Design formulations are derived from a parametric study carried out through numerical simulations of several geometric configurations.

## 1. Introduction

One of the most common ways to design seismic-resistant steel structures is by adopting Moment Resisting Frames (MRFs). They are systems conceived to withstand horizontal loads through the flexural behaviour of structural members (i.e., beams, columns and connections). As highlighted by Astaneh [1], several typologies of MRFs can be individuated in practice, differing for the three-dimensional layout, the connection typology, the ductility class, the selection of the dissipative zone.

Concerning the spatial distribution, MRFs can be classified as space frames and perimeter frames. In the first case, 3D frames withstand vertical and horizontal loads, which is very effective from the structural point of view because it reduces the beam and column sizes, limiting, at the same time, the forces in the nodal components (welds and bolts). Nevertheless, this approach requires a significant manufacturing effort since several complex beam-to-column connections have to be fabricated. Conversely, perimeter-framed buildings are characterised by MRFs located on the sides, while the remaining frames exhibit pendular behaviour. In this case, only few connections are designed to sustain the horizontal loads with low manufacturing costs. Nevertheless, typically deeper beams and columns are required.

Although technical requirements can inspire the choice of adopting one of the MRF typologies previously summarised, the selection also depends on the construction tradition of different countries. For instance, one-way perimeter MRFs are widely exploited in Europe and the United States due to their ability to clearly identify and design the seismic and gravity frames and limit the use of expensive joints only in MRFs. Conversely, space frames are preferred, for instance, in the Japanese tradition. The relevant consequence of this choice is the need to select circular or square tubular columns since they are able to provide strength and stiffness along both the main axes. Even though this solution requires a relevant number of joints, it has the beneficial effect of optimising the design of the beams limiting their depth and, consequently, the oversizing of brittle nodal components. Nevertheless, the main drawback of this practice is the need to manufacture complex beam-to-column connections [2,3,4,5,6,7,8,9,10], usually characterised by the use of collar plates and stiffeners.

In this framework, the recent introduction of 3D Laser Cutting Technology (3D-LCT) in Civil Engineering provides a way to simplify the fabrication of complex configurations and exploit tubular columns more easily in the European construction market. An example of this simplification is the possibility to easily realise Circular Hollow Section (CHS) columns with passing-through double-tee beam connections [11,12] (Figure 1). However, the novelty of this solution represents a limit to its application in practice because no design equations are included in common standards. In fact, the current draft of Eurocode 8 does not provide design equations for these typologies or extends formulas which result in being overconservative [11,12,13].

Recently, research works have started to fill this knowledge gap. For instance, the LASTEICON research project [14,15,16,17], financed by the European Commission, has studied the technical aspects related to the use of the 3D-LCT in Civil Engineering. Furthermore, attention has been devoted to defining design equations for exploiting the CHS column and through I-beam joints in the construction market. In order to complement this work, some studies are ongoing at the University of Salerno. In particular, experimental, numerical and theoretical activities have been carried out to extend the component method to these typologies [18,19,20,21]. With this goal, the main components of this joint have been identified in [22] (Figure 2):

pct/pcc are the components representative of the behaviour in tension and compression of the attachment between the tubular profile and the flange plates of the beam;

ttt/ttc are sources of deformability ascribed to the possible transverse tension or compression actions transmitted by the flanges to the CHS and induced by the beam’s rotation;

cs and bws represent the parts of the tubular profile and the beam web subjected to the horizontal shear.

Figure 2 shows the identified components and the parallel and series links which allow the mechanical modelling of the whole joint. Similarly to EC3 part 1.8, all the components have been modelled through elastic perfectly-plastic laws. In particular, the sources of deformability pcc/pct have been studied in [22] by characterising both strength and stiffness and proposing design formulations shown in Equations (1) and (2).
(1)$$F_{pcc/pct}=0.24\beta^{-0.41}\gamma^{0.43}\tau^{0.47}\frac{5.98}{1-\beta^{0.52}}f_{y}t_{0}^{2},$$
(2)$$k_{pcc/pct}=9.6\frac{\beta^{-1.17}\gamma^{-2.42}\tau^{1.31}}{\beta^{-2.61}-1}Ed_{0}.$$

In Equations (1) and (2) $\beta =\frac{b_{1}}{d_{0}}$, $\gamma =\frac{d_{0}}{2t_{0}}$ and $\tau =\frac{t_{1}}{t_{0}}$ where $b_{1}$ and $t_{1}$ are the plate’s width and thickness, while $d_{0}$ and $t_{0}$ are the CHS diameter and thickness, respectively.

The shear components cs and bws have been characterised by extending the same formulas already proposed by Eurocode 3 part 1.8 [5]:(3)$$F_{cs}=\frac{0.9A_{v,cs}f_{y}}{\sqrt{3}\gamma_{M0}\beta_{v}},$$
(4)$$k_{cs}=\frac{0.38EA_{v,cs}}{\beta_{v}z},$$
(5)$$F_{bws}=\frac{0.9A_{v,bws}f_{y}}{\sqrt{3}\gamma_{M0}\beta_{v}},$$
(6)$$k_{bws}=\frac{0.38EA_{v,bws}}{\beta_{v}z},$$
where in Equations (3)–(6), $A_{v,cs}$ is the shear area of the tubular profile ($A_{v,cs}=2A/\pi$), $A_{v,bws}$ is the shear area of the beam web ($A_{v,bws}=d_{0}t_{bw}$), $t_{bw}$ is the web thickness of the beam, $\gamma_{M0}=1$ is the partial safety factor, $f_{y}$ is the yield stress, $\beta_{v}$ is the transformation parameter equal to $\beta_{v}=1-\frac{z}{L_{c}}$, z is the lever arm (distance between the centerlines of the beam flanges) and $L_{c}$ is the column length.

Consequently, to achieve a complete knowledge of the whole joint behaviour between CHS columns and passing through I-beams with the component method, only the study of the sources of deformability ttt/ttc (Figure 3) is still needed. Therefore, the main objective of this paper is the characterisation of this component which is representative of the column parts which are subjected locally to transverse tension or compression actions transmitted by the beam flanges.

The paper is structured as follows: in paragraph 2, the methodology of the work is briefly reported, focusing on the numerical implementation of the analysed connection in Abaqus [23]. The current investigation intends to define strength and stiffness formulations of the examined component through a parametric analysis of 31 different geometrical configurations of connections whose behaviour is numerically simulated. Paragraph 3 concentrates on the main results of the simulations and provides the required strength and stiffness equations of the component employing regression analyses of results obtained from the parametric study.

## 2. Materials and Methods

The main aim of the current work is the strength and stiffness characterisation of the components ttt/ttc belonging to CHS to passing-through double-tee beam connection, representing the possible transverse tension/compression on the tubular section generated by the rotation of the beam flanges. The need to investigate these components derives from preliminary monotonic and cyclic tests performed on beam-to-column sub-assemblies at the University of Salerno [11], highlighting that the column parts close to the flange-to-column attachment may experience buckling or tearing phenomena if they are respectively loaded in compression or tension (Figure 3).

The study of ttt/ttc components was carried out through a parametric analysis concerning 31 geometrical configurations of specimens representative of the connections between CHS columns and flanges of double-tee profiles. The mechanical model of the considered configurations refers to an axially loaded cantilever scheme shown in Figure 4. In particular, to study the behaviour of the analysed components in tension and compression, only the sides of the plate close to the fixed constraint were welded to the column; instead, contact properties were applied for the remaining sides indicated as Free sides in Figure 4.

Starting from the results provided in [11], it was possible to observe that the main parameters affecting the response of the analysed component are β and γ. Consequently, the 31 cases were appropriately selected by varying β between 0.44 and 0.72, with γ ranges between 13.69 and 39.51 (Table 1).

The analyses were conducted using the Finite Element (FE) software Abaqus 6.17 [23]. In particular, the adopted scheme refers to the specimens tested in [11], which are characterised by a CHS profile with a length equal to 500 mm, restrained at one end with a fixed constraint while the other end is free. The hole to pass through the plate is modelled without tolerance, and Tie constraints ensure the connection between the plate and the tube to avoid modelling the welds. The material properties associated with the plate and the CHS refer to a nominal S355JR steel modelled with a quadri-linear constitutive law, as proposed by Faella et al. [24], with an elastic modulus equal to 210 GPa and a Poisson’s ratio of 0.30. The damage evolution of the material properties was included in the analysis by implementing an equivalent plastic strain at fracture equal to 2.4 mm, complying with the recommendations provided by [25,26]. The plates and the tubular profiles were meshed by adopting C3D8-type (8-node linear brick) elements with a 5 mm size. The simulations were carried out employing a static solver and applying increasing displacements along the plate face in the direction of the longitudinal axis of the column, as reported in Figure 5.

The developed FE model is the same one validated in [22] against the results of experimental activities carried out on the components pcc/pct, with the only difference in the way of application of the displacement condition; consequently, no repetitive information is provided in this manuscript.

It is worth highlighting that referring to the component in compression, the monotonic simulation was anticipated by a buckling analysis intended to define the shape of the imperfection to be applied to the specimen. Therefore, the buckling mode was amplified according to the requirements of Eurocode 3 part 1.5 [27] and EN 10034 [28] with a scale factor reproducing an imperfection on the tube equal to the 3% of the nominal internal diameter (Figure 6).

The choice of applying displacements along the longitudinal direction only without also the rotation of the plate is ascribed to the mechanical behaviour observed during the experimental campaigns carried out in [11]. In fact, as it has also been confirmed by numerical simulations (Figure 7) that the part of the passing-through plate in the tubular profile is connected to the other nodal components and acts as a rigid plate; consequently, the actions transmitted by the beam become loads acting orthogonally to the plate. Moreover, the rotation of the flanges in the column is lower than the rotation exhibited at the beam-to-column attachment, justifying the fact that this effect can be ignored.

## 3. Results and Discussion

This section discusses the numerical results of the parametric analysis mentioned above. Monotonic tests were carried out, and the values of stiffness and resistance were defined. In particular, the resistance of the analysed component was evaluated when the first element of the model achieved a principal plastic tensile deformation equal to the 5%. Conversely, the stiffness was assessed when the force experienced by the connections was equal to 2/3 the plastic resistance. The main outcomes are summarised in Table 2, where the strength in compression ($F_{ttc}$), tension ($F_{ttt}$) and stiffness ($k_{ttt}=k_{ttc}=k_{ttt/ttc}$) are reported.

As an example, images concerning the von Mises stress and principal plastic deformation distributions for Case 10 are shown in Figure 8, Figure 9, Figure 10 and Figure 11.

### 3.1. CHS Tube under Localised Transverse Compression (ttc)

It is worth highlighting that the specimens were adequately selected to define sets of cases characterised by the same values of β and γ. Consequently, the grouping of cases with the same characteristics (Figure 12) allowed to highlight that the strength under localised transverse compression depends on parameter β according to an exponential law with power varying between 0.56 and 0.80. Similarly, for the parameter γ the exponent varies between −1.11 and −1.16. 

To define a formula for the prediction of the resistance of this component, a regression study was performed referring to the dimensionless parameters $F_{ttc}/\left(b_{1}t_{0}f_{y}\right)$ versus β and γ. The regression analysis led to Equation (7).
(7)$$F_{ttc}=\beta^{0.46}\gamma^{0.2}b_{1}t_{0}f_{y}.$$

The reliability of the proposed formulation is demonstrated in Table 3 through the comparison between the numerical outcomes and the analytical proposal. In most cases, the prediction is accurate, with a coefficient of variation equal to 10.4%.

### 3.2. CHS Tube under Localised Transverse Tension (ttt)

The same approach was applied to characterise the resistance of the tube under localised transverse tension, $F_{ttt}$.

The strength of this component depends on β according to an exponential law with power varying between 0.82 and 0.91, while referring to γ, the power is about –1.02 (Figure 13). For this reason, a regression analysis was performed considering the dimensionless parameters $F_{ttt}/\left(b_{1}t_{0}f_{y}\right)$, β and γ, obtaining Equation (8).
(8)$$F_{ttt}=\beta^{0.12}\gamma^{0.16}b_{1}t_{0}f_{y}.$$

The accuracy of the proposed formulation is demonstrated against the results of the numerical simulations, as shown in Table 4. In most cases, the prediction is accurate, even though a coefficient of variation equal to 6.6% was observed.

### 3.3. Stiffness Formulation

Considering that the equations adopted to define the resistance of the analysed components are reported in the simple formulation $F/b_{1}t_{0}f_{y}=\beta^{constant1}\gamma^{constant2}$, it was chosen to adapt this equation also to characterise the stiffness (k) according to Equation (9):(9)$$k/\left(b_{1}E\right)=\beta^{constant1}\gamma^{constant2}.$$

A regression study was carried out starting from the numerical outcomes of the parametric analysis, leading to Equation (10):(10)$$k=\beta^{0.22}\gamma^{-0.80}b_{1}E.$$

The accuracy of the derived equation is proven in Table 5, where it is shown that the mean of the ratios between predicted results and FE outcomes is equal to 1.00, while the coefficient of variation is about 14.4%. This equation is to be considered valid both for the case of compression and tension.

## 4. Conclusions

The present work characterised the mechanical modelling of the stiffness and strength of two of the main components of joints with Circular Hollow Section columns and passing-through double-tee beams. In particular, the analysed sources of deformability are activated by the rotation of the beam flanges and generate local compressive (ttc) and tensile (ttt) actions on the tubular profile. The first step of the study consisted in a proper selection of 31 geometrical configurations of CHS columns with passing-through plates whose behaviour was assessed with numerical simulations with Finite Element software. Monotonic simulations were carried out imposing an increasing displacement of the plates in the direction of the tubular axis. For each analysed case, the strength and stiffness were derived. The results showed that, generally, the strength and stiffness of the analysed components have an exponential dependence on the parameters β and γ. Consequently, regression analyses were carried out to propose design formulations which proved their accuracy since the scatters with the numerical outcomes are characterised by coefficients of variation equal to 10.4%, 6.6% and 14.4% in terms of strength in compression, tension and stiffness, respectively.

Future developments of the current research will consist in collecting information about all the nodal components involved in characterising the flexural strength of CHS columns and passing-through I-beams so that a component modelling of the joint can be proposed.

## Figures and Tables

**Figure 1 materials-16-02641-f001:**
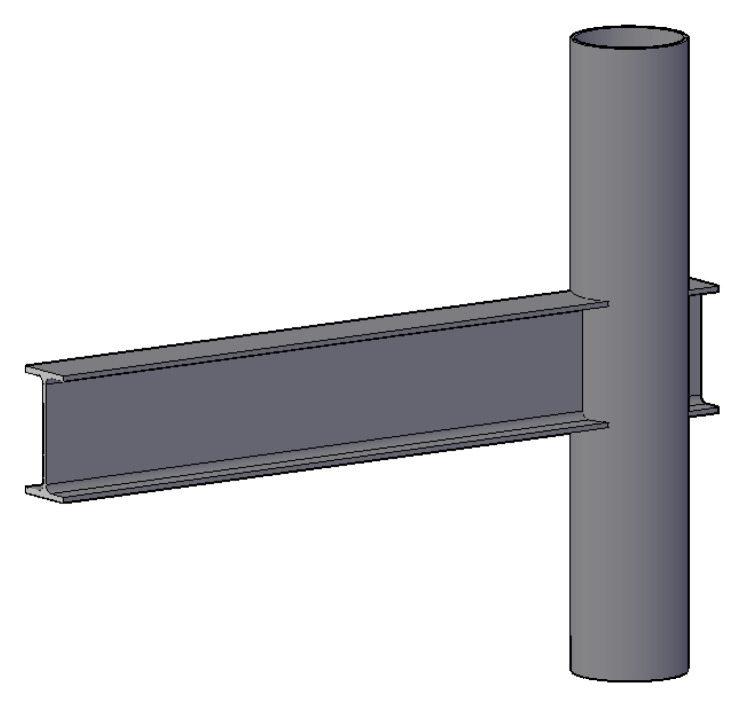
Connection between CHS column and passing-through double-tee beam.

**Figure 2 materials-16-02641-f002:**
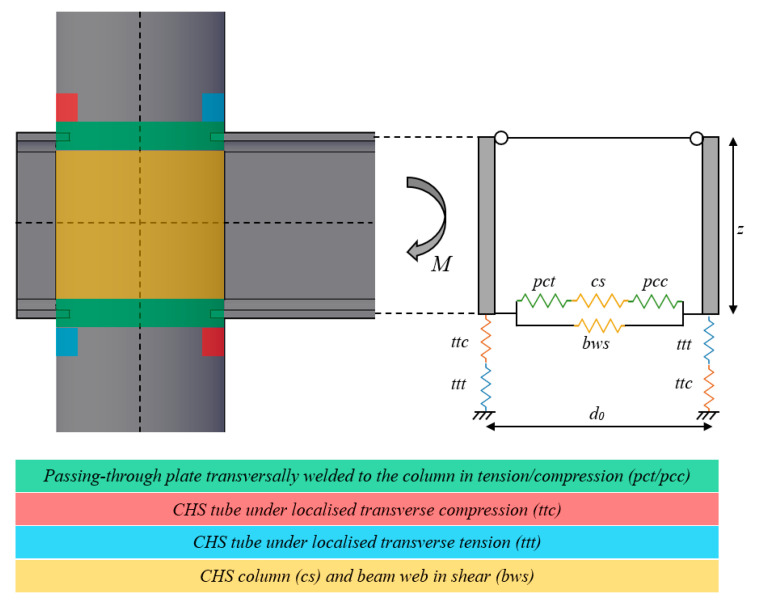
Selection of the nodal components according to the component method approach.

**Figure 3 materials-16-02641-f003:**
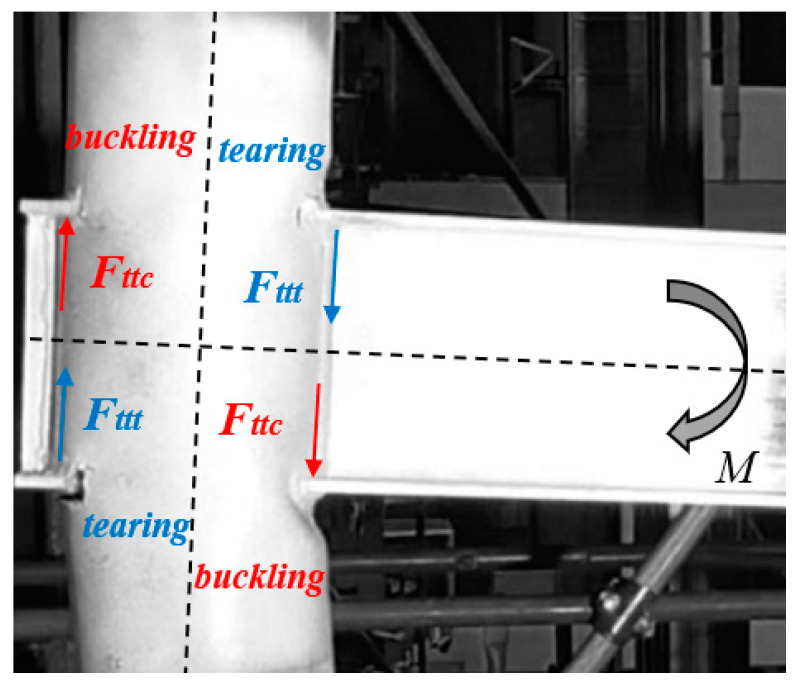
Actions applied to the components ttt/ttc induced by the rigid rotation of the beam.

**Figure 4 materials-16-02641-f004:**
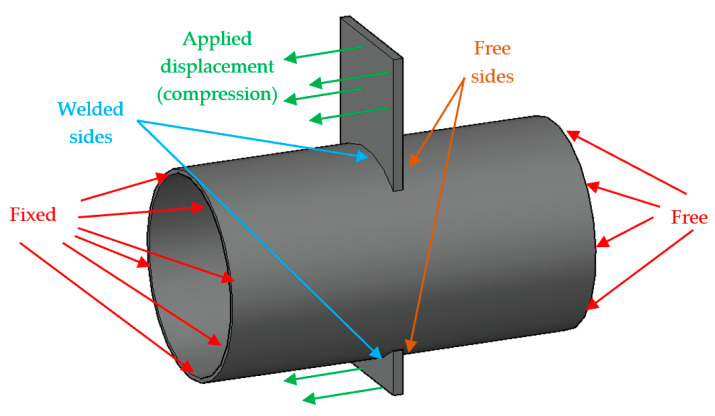
Mechanical model of the analysed configurations.

**Figure 5 materials-16-02641-f005:**
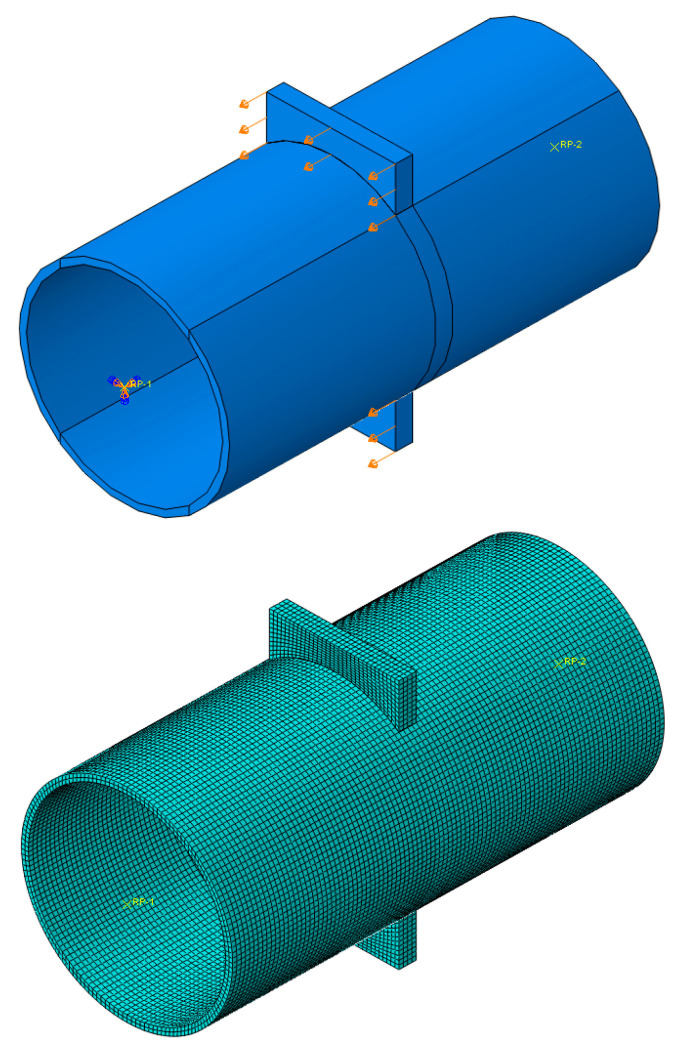
Detail of the loaded part of the connection (**top**) and of the mesh (**bottom**).

**Figure 6 materials-16-02641-f006:**
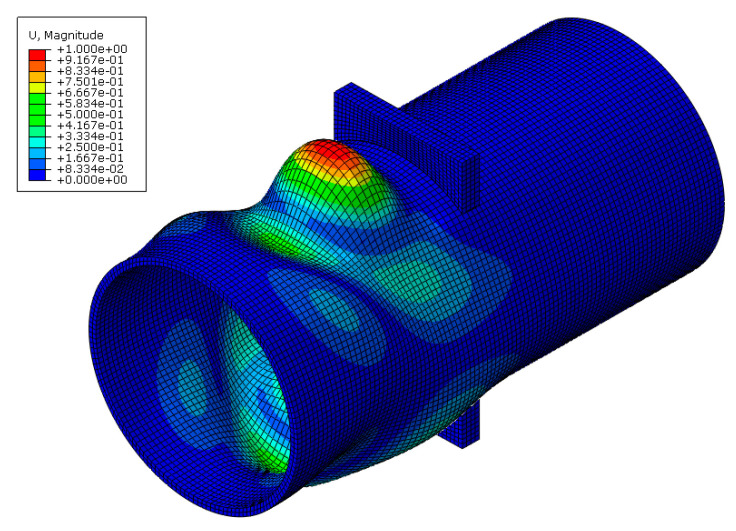
Considered buckling mode.

**Figure 7 materials-16-02641-f007:**
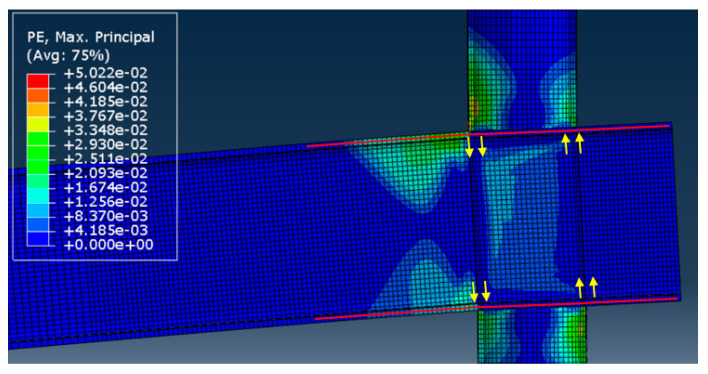
View-cut of a CHS column with passing-through beam connection.

**Figure 8 materials-16-02641-f008:**
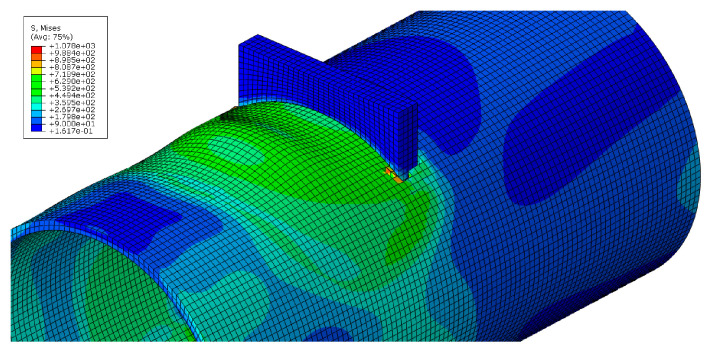
von Mises stress distribution of the tube under localised transverse compression (3D view, Case 10).

**Figure 9 materials-16-02641-f009:**
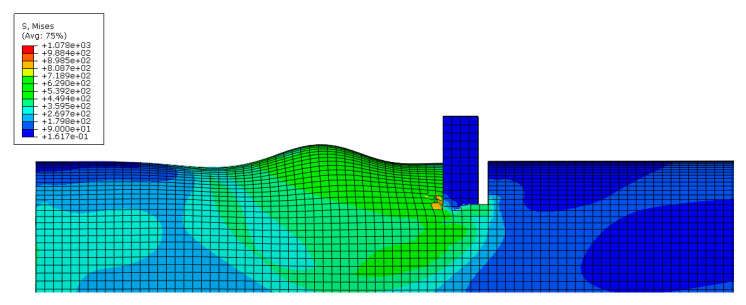
von Mises stress distribution of the tube under localised transverse compression (lateral view, Case 10).

**Figure 10 materials-16-02641-f010:**
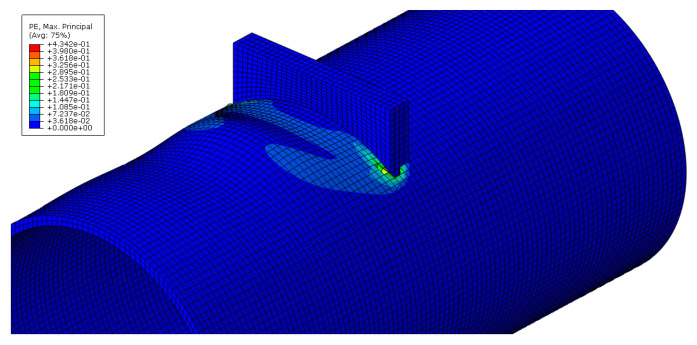
Principal plastic deformation of the tube under localised transverse compression (3D view, Case 10).

**Figure 11 materials-16-02641-f011:**
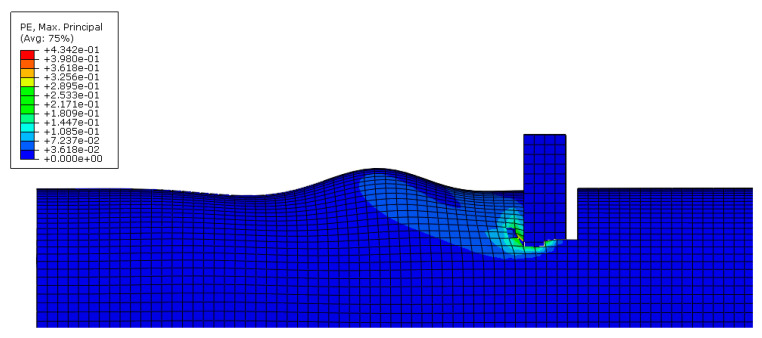
Principal plastic deformation of the tube under localised transverse compression (lateral view, Case 10).

**Figure 12 materials-16-02641-f012:**
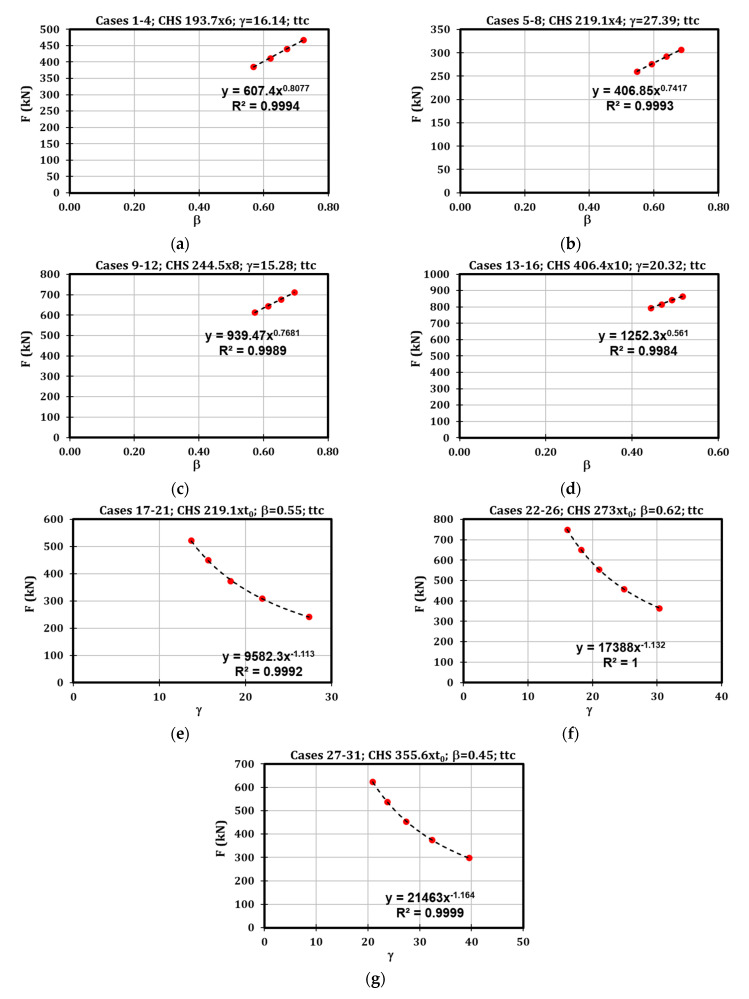
Results of the parametric analysis referred to $F_{ttc}$.

**Figure 13 materials-16-02641-f013:**
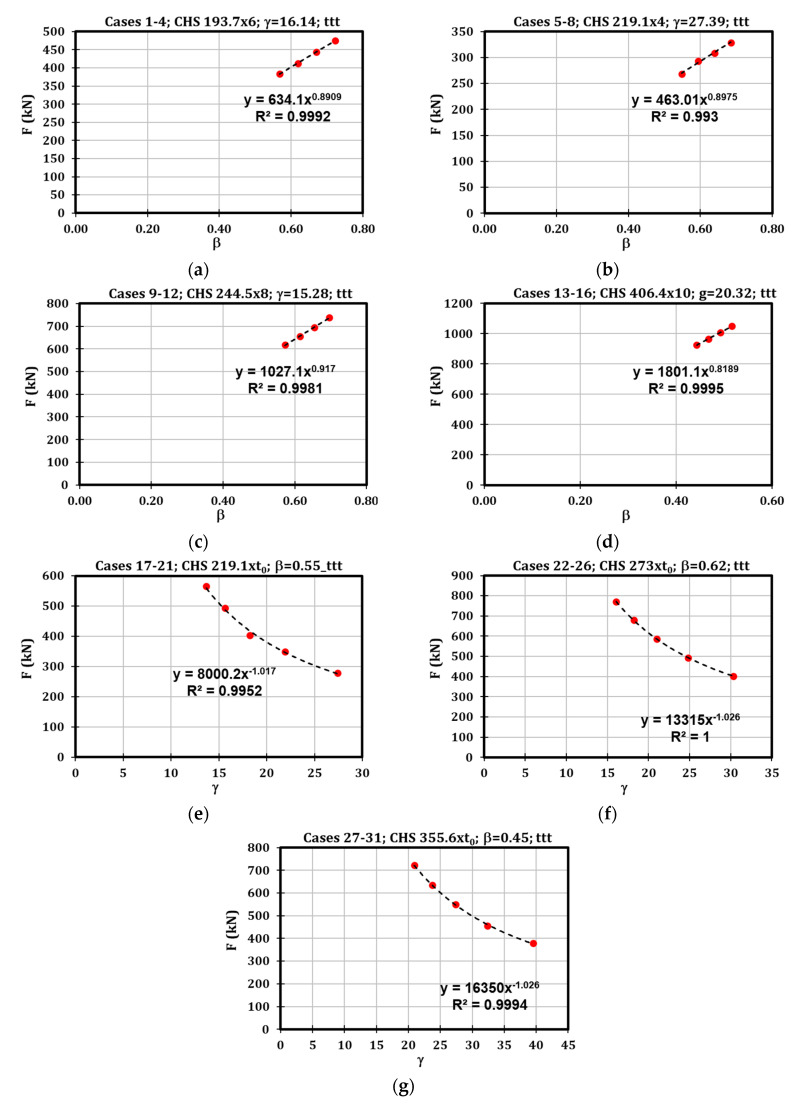
Results of the parametric analysis referring to $F_{ttt}$.

**Table 1 materials-16-02641-t001:** Selected geometrical configurations for the parametric analysis.

Case	Column Diameter d_0_ (mm)	Column Thickness t_0_ (mm)	Plate Width b_1_ (mm)	β	γ
1	193.7	6	110	0.57	16.14
2	193.7	6	120	0.62	16.14
3	193.7	6	130	0.67	16.14
4	193.7	6	140	0.72	16.14
5	219.1	4	120	0.55	27.39
6	219.1	4	130	0.59	27.39
7	219.1	4	140	0.64	27.39
8	219.1	4	150	0.68	27.39
9	244.5	8	140	0.57	15.28
10	244.5	8	150	0.61	15.28
11	244.5	8	160	0.65	15.28
12	244.5	8	170	0.70	15.28
13	406.4	10	180	0.44	20.32
14	406.4	10	190	0.47	20.32
15	406.4	10	200	0.49	20.32
16	406.4	10	210	0.52	20.32
17	219.1	4	120	0.55	27.39
18	219.1	5	120	0.55	21.91
19	219.1	6	120	0.55	18.26
20	219.1	7	120	0.55	15.65
21	219.1	8	120	0.55	13.69
22	273	4.5	170	0.62	30.33
23	273	5.5	170	0.62	24.82
24	273	6.5	170	0.62	21.00
25	273	7.5	170	0.62	18.20
26	273	8.5	170	0.62	16.06
27	355.6	4.5	160	0.45	39.51
28	355.6	5.5	160	0.45	32.33
29	355.6	6.5	160	0.45	27.35
30	355.6	7.5	160	0.45	23.71
31	355.6	8.5	160	0.45	20.92

**Table 2 materials-16-02641-t002:** Results of the numerical simulations.

Case	d_0_ (mm)	t_0_ (mm)	b_1_ (mm)	β	γ	F_ttc_ (kN)	F_ttt_ (kN)	k_ttt/ttc_ (N/mm)
1	193.7	6	110	0.57	16.14	385	384	2,669,973
2	193.7	6	120	0.62	16.14	411	412	2,836,234
3	193.7	6	130	0.67	16.14	440	444	3,040,961
4	193.7	6	140	0.72	16.14	468	476	3,278,204
5	219.1	4	120	0.55	27.39	260	268	1,521,235
6	219.1	4	130	0.59	27.39	276	293	1,609,283
7	219.1	4	140	0.64	27.39	293	309	1,703,291
8	219.1	4	150	0.68	27.39	307	329	1,769,345
9	244.5	8	140	0.57	15.28	613	618	3,383,171
10	244.5	8	150	0.61	15.28	645	654	3,560,892
11	244.5	8	160	0.65	15.28	677	694	3,757,766
12	244.5	8	170	0.70	15.28	712	738	4,015,597
13	406.4	10	180	0.44	20.32	793	926	2,909,890
14	406.4	10	190	0.47	20.32	816	965	2,931,171
15	406.4	10	200	0.49	20.32	843	1008	2,944,675
16	406.4	10	210	0.52	20.32	864	1049	2,907,151
17	219.1	4	120	0.55	27.39	242	279	2,192,166
18	219.1	5	120	0.55	21.91	309	349	1,872,032
19	219.1	6	120	0.55	18.26	373	404	2,168,739
20	219.1	7	120	0.55	15.65	451	493	2,859,602
21	219.1	8	120	0.55	13.69	523	565	3,371,274
22	273	4.5	170	0.62	30.33	365	403	1,617,685
23	273	5.5	170	0.62	24.82	458	493	2,076,532
24	273	6.5	170	0.62	21.00	554	587	2,568,008
25	273	7.5	170	0.62	18.20	651	679	3,070,647
26	273	8.5	170	0.62	16.06	750	772	3,588,121
27	355.6	4.5	160	0.45	39.51	298	379	1,103,192
28	355.6	5.5	160	0.45	32.33	375	456	1,399,880
29	355.6	6.5	160	0.45	27.35	454	550	1,736,309
30	355.6	7.5	160	0.45	23.71	539	636	2,130,192
31	355.6	8.5	160	0.45	20.92	625	723	2,529,651

**Table 3 materials-16-02641-t003:** Validation of the proposed formulation related to the strength in compression.

Test	d_0_ (mm)	t_0_ (mm)	b_1_ (mm)	β	γ	F_ttc, FEM_ (kN)	F_ttc, proposal_ (kN)	F_ttc, proposal_/F_ttc, FEM_
1	193.7	6	110	0.57	16.14	385	312	0.81
2	193.7	6	120	0.62	16.14	411	354	0.86
3	193.7	6	130	0.67	16.14	440	398	0.90
4	193.7	6	140	0.72	16.14	468	444	0.95
5	219.1	4	120	0.55	27.39	260	248	0.95
6	219.1	4	130	0.59	27.39	276	278	1.01
7	219.1	4	140	0.64	27.39	293	310	1.06
8	219.1	4	150	0.68	27.39	307	343	1.12
9	244.5	8	140	0.57	15.28	613	526	0.86
10	244.5	8	150	0.61	15.28	645	582	0.90
11	244.5	8	160	0.65	15.28	677	639	0.94
12	244.5	8	170	0.70	15.28	712	698	0.98
13	406.4	10	180	0.44	20.32	793	795	1.00
14	406.4	10	190	0.47	20.32	816	860	1.05
15	406.4	10	200	0.49	20.32	843	927	1.10
16	406.4	10	210	0.52	20.32	864	995	1.15
17	219.1	4	120	0.55	27.39	242	248	1.02
18	219.1	5	120	0.55	21.91	309	296	0.96
19	219.1	6	120	0.55	18.26	373	343	0.92
20	219.1	7	120	0.55	15.65	451	388	0.86
21	219.1	8	120	0.55	13.69	523	432	0.83
22	273	4.5	170	0.62	30.33	365	427	1.17
23	273	5.5	170	0.62	24.82	458	502	1.10
24	273	6.5	170	0.62	21.00	554	574	1.04
25	273	7.5	170	0.62	18.20	651	644	0.99
26	273	8.5	170	0.62	16.06	750	712	0.95
27	355.6	4.5	160	0.45	39.51	298	365	1.23
28	355.6	5.5	160	0.45	32.33	375	429	1.14
29	355.6	6.5	160	0.45	27.35	454	490	1.08
30	355.6	7.5	160	0.45	23.71	539	550	1.02
31	355.6	8.5	160	0.45	20.92	625	608	0.97
					**Mean**			1.00
					**Standard deviation**			0.104
					**Coefficient of variation**			0.104

**Table 4 materials-16-02641-t004:** Validation of the proposed formulation related to the strength in tension.

Test	d_0_ (mm)	t_0_ (mm)	b_1_ (mm)	β	γ	F_ttt, FEM_ (kN)	F_ttt, proposal_ (kN)	F_ttt, proposal_/F_ttt, FEM_
1	193.7	6	110	0.57	16.14	384	344	0.90
2	193.7	6	120	0.62	16.14	412	379	0.92
3	193.7	6	130	0.67	16.14	444	415	0.93
4	193.7	6	140	0.72	16.14	476	451	0.95
5	219.1	4	120	0.55	27.39	268	271	1.01
6	219.1	4	130	0.59	27.39	293	297	1.01
7	219.1	4	140	0.64	27.39	309	323	1.04
8	219.1	4	150	0.68	27.39	329	349	1.06
9	244.5	8	140	0.57	15.28	618	579	0.94
10	244.5	8	150	0.61	15.28	654	626	0.96
11	244.5	8	160	0.65	15.28	694	673	0.97
12	244.5	8	170	0.70	15.28	738	720	0.98
13	406.4	10	180	0.44	20.32	926	945	1.02
14	406.4	10	190	0.47	20.32	965	1004	1.04
15	406.4	10	200	0.49	20.32	1008	1064	1.05
16	406.4	10	210	0.52	20.32	1049	1123	1.07
17	219.1	4	120	0.55	27.39	279	271	0.97
18	219.1	5	120	0.55	21.91	349	327	0.94
19	219.1	6	120	0.55	18.26	404	381	0.94
20	219.1	7	120	0.55	15.65	493	434	0.88
21	219.1	8	120	0.55	13.69	565	485	0.86
22	273	4.5	170	0.62	30.33	403	447	1.11
23	273	5.5	170	0.62	24.82	493	528	1.07
24	273	6.5	170	0.62	21.00	587	608	1.04
25	273	7.5	170	0.62	18.20	679	685	1.01
26	273	8.5	170	0.62	16.06	772	761	0.99
27	355.6	4.5	160	0.45	39.51	379	422	1.11
28	355.6	5.5	160	0.45	32.33	456	499	1.09
29	355.6	6.5	160	0.45	27.35	550	574	1.04
30	355.6	7.5	160	0.45	23.71	636	647	1.02
31	355.6	8.5	160	0.45	20.92	723	719	0.99
					**Mean**			1.00
					**Standard deviation**			0.066
					**Coefficient of variation**			0.066

**Table 5 materials-16-02641-t005:** Validation of the proposed formulation related to the stiffness.

Test	d_0_ (mm)	t_0_ (mm)	b_1_ (mm)	β	γ	k_FEM_ (N/mm)	k_proposal_ (N/mm)	k_proposal_/k_FEM_
1	193.7	6	110	0.57	16.14	2,669,973	2,176,253	0.82
2	193.7	6	120	0.62	16.14	2,836,234	2,420,718	0.85
3	193.7	6	130	0.67	16.14	3,040,961	2,669,784	0.88
4	193.7	6	140	0.72	16.14	3,278,204	2,923,173	0.89
5	219.1	4	120	0.55	27.39	1,521,235	1,539,651	1.01
6	219.1	4	130	0.59	27.39	1,609,283	1,698,065	1.06
7	219.1	4	140	0.64	27.39	1,703,291	1,859,228	1.09
8	219.1	4	150	0.68	27.39	1,769,345	2,022,986	1.14
9	244.5	8	140	0.57	15.28	3,383,171	2,899,806	0.86
10	244.5	8	150	0.61	15.28	3,560,892	3,155,217	0.89
11	244.5	8	160	0.65	15.28	3,757,766	3,414,466	0.91
12	244.5	8	170	0.70	15.28	4,015,597	3,677,363	0.92
13	406.4	10	180	0.44	20.32	2,909,890	2,799,625	0.96
14	406.4	10	190	0.47	20.32	2,931,171	2,991,089	1.02
15	406.4	10	200	0.49	20.32	2,944,675	3,184,819	1.08
16	406.4	10	210	0.52	20.32	2,907,151	3,380,727	1.16
17	219.1	4	120	0.55	27.39	2,192,166	1,539,651	0.70
18	219.1	5	120	0.55	21.91	1,872,032	1,842,133	0.98
19	219.1	6	120	0.55	18.26	2,168,739	2,132,891	0.98
20	219.1	7	120	0.55	15.65	2,859,602	2,414,249	0.84
21	219.1	8	120	0.55	13.69	3,371,274	2,687,802	0.80
22	273	4.5	170	0.62	30.33	1,617,685	2,067,695	1.28
23	273	5.5	170	0.62	24.82	2,076,532	2,429,628	1.17
24	273	6.5	170	0.62	21.00	2,568,008	2,778,803	1.08
25	273	7.5	170	0.62	18.20	3,070,647	3,117,552	1.02
26	273	8.5	170	0.62	16.06	3,588,121	3,447,527	0.96
27	355.6	4.5	160	0.45	39.51	1,103,192	1,463,307	1.33
28	355.6	5.5	160	0.45	32.33	1,399,880	1,719,447	1.23
29	355.6	6.5	160	0.45	27.35	1,736,309	1,966,558	1.13
30	355.6	7.5	160	0.45	23.71	2,130,192	2,206,290	1.04
31	355.6	8.5	160	0.45	20.92	2,529,651	2,439,813	0.96
					**Mean**			1.00
					**Standard deviation**			0.144
					**Coefficient of variation**			0.144

## Data Availability

Not available.

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
