# Peer review of "Mechanical Modelling of the Strength and Stiffness of Circular Hollow Section Tube under Localised Transverse Compression and Tension"

_materials, 2023, doi:10.3390/ma16072641_

Round 1

Reviewer 1 Report

It is an interesting approach of modelling regarding the strength and stiffness of circular-hollow-section tube under localised transverse compression and tension. The manuscript totally lacks the experimental part which was previously published in references [11, 12, 20]. However, the presented model might be of interest for the specialists in the field. Therefore the manuscript title must be properly adapted and several manuscript improvements are required by considering the comments below:

Comment 1) Left side of Figure 2 is identical with Figure 6 in reference [20], it is noteworthy that Figure 2 caption contains the citation to the reference [20]. Figure 3 is assembled by parts of Figures 7 and 8 from the reference [20] but no reference is given in the caption. The copyright of reference [20] belongs to the Elsevier Publishing House (not to you as authors of the article), therefore need Elsevier written permission to reprint parts o the figures. However, such situation is acceptable for review type article but not for a research article. So, it is recommended to remove Figure 2 and 3 from yours current manuscript or to present the Elsevier written permission to reprint these figures.

Comment 2) The manuscript title in current form mislead the readers and therefore must be changed to give a proper description of the article subject such as: ,,Mechanical modelling of the strength and stiffness of Circular-Hollow-Section tube under localised transverse compression and tension

Comment 3) Lines 236 – 237 the text: ,,The present work has characterised the stiffness and strength of two of the main components of joints with Circular-Hollow-Section…” should be modified according to the manuscript subject in the following manner ,,The present work has characterised the mechanical modelling of the stiffness and strength of two of the main components of joints with Circular-Hollow-Section…”

Comment 4) The current state of the work in the field is based almost on the old references. The newest references cited in the manuscript are yours self citation [11, 12, 20], fortunately they are adequate and properly placed in the context. So, the manuscript must be improved by adding some newest references (2020 – 2023) in the field to the work published by the others researchers and compare their results with those obtained by yours proposed model.

Author Response

Comment 1

Left side of Figure 2 is identical with Figure 6 in reference [20], it is noteworthy that Figure 2 caption contains the citation to the reference [20]. Figure 3 is assembled by parts of Figures 7 and 8 from the reference [20] but no reference is given in the caption. The copyright of reference [20] belongs to the Elsevier Publishing House (not to you as authors of the article), therefore need Elsevier written permission to reprint parts o the figures. However, such situation is acceptable for review type article but not for a research article. So, it is recommended to remove Figure 2 and 3 from yours current manuscript or to present the Elsevier written permission to reprint these figures.

Response 1

Figures 2 and 3 have been substituted, as suggested by the reviewer.

Comment 2

The manuscript title in current form mislead the readers and therefore must be changed to give a proper description of the article subject such as: “Mechanical modelling of the strength and stiffness of Circular-Hollow-Section tube under localised transverse compression and tension”

Response 2

The title has been modified according to the reviewer’s suggestion.

Comment 3

Lines 236 – 237 the text: “The present work has characterised the stiffness and strength of two of the main components of joints with Circular-Hollow-Section…” should be modified according to the manuscript subject in the following manner “The present work has characterised the mechanical modelling of the stiffness and strength of two of the main components of joints with Circular-Hollow-Section…”

Response 3

Lines 236-237 have been modified according to the reviewer’s suggestion.

Comment 4

The current state of the work in the field is based almost on the old references. The newest references cited in the manuscript are yours self citation [11, 12, 20], fortunately they are adequate and properly placed in the context. So, the manuscript must be improved by adding some newest references (2020 – 2023) in the field to the work published by the others researchers and compare their results with those obtained by yours proposed model.

Response 4

The components analysed in this paper (ttc/ttt) represent a novelty, and, at the moment, no references are available. Consequently, no comparison can be provided.

However, the future developments of the work are perfectly in line with the reviewer’s suggestion. The authors will assemble all the components studied at the moment and propose a model which will need to be validated against the formulations proposed by the participants of the LASTEICON research project.

Even though a comparison cannot be performed, additional references have been added referring to the results of the LASTEICON research project, as suggested by the reviewer:

- Das, R., Castiglioni, C.A., Couchaux, M., Hoffmeister, B., Degee, H., Design and analysis of laser-cut based moment resisting passing-through I-beam-to-CHS column joints, Journal of Constructional Steel Research, Volume 169, 106015, ISSN 0143-974X, https://doi.org/10.1016/j.jcsr.2020.106015, 2020.

- Couchaux, M., Vyhlas, V., Kanyilmaz, A., Hjiaj, M., Passing-through I-beam-to-CHS column joints made by laser cutting technology: Experimental tests and design model, Journal of Constructional Steel Research, 176, art. no. 106298, DOI: 10.1016/j.jcsr.2020.106298, 2021.

Reviewer 2 Report

Materials journal focuses on materials, their properties, materials diagnostics, .... The article is about the design of structural elements and is, in my opinion, outside the focus of the magazine. The article describes simulation results of specific structural elements and belongs to the field of strength analysis, not to the field of materials. It is written in the form of a technical report. The article is an engineering solution, not a scientific article.
Apart from the above, there is nothing to assess in the article. The methodology of creating a numerical model is not investigated, yet modelling the connection using a weld would give scope for research: e.g., a simplified procedure for modelling a connection without a weld could be proposed based on experimental measurements... The material model is chosen without deeper discussion and without verification. The mesh is simple, regular. The contents are simulations of simple geometric shapes for the nonlinear material model. The simulation tool used - Abaqus - is a commonly available software. The structural node model used in the paper is not difficult and requires only the usual engineering skills. There is nothing scientific or new in the article. The reason is that this is not a scientific article but a technical report on strength analyses.
In terms of terminology - principal plastic deformation is principal plastic strain. It would be interesting to know its direction as well. However, the stresses are only given by von Mises... 

Author Response

Comment

Materials journal focuses on materials, their properties, materials diagnostics, .... The article is about the design of structural elements and is, in my opinion, outside the focus of the magazine. The article describes simulation results of specific structural elements and belongs to the field of strength analysis, not to the field of materials. It is written in the form of a technical report. The article is an engineering solution, not a scientific article.

Apart from the above, there is nothing to assess in the article. The methodology of creating a numerical model is not investigated, yet modelling the connection using a weld would give scope for research: e.g., a simplified procedure for modelling a connection without a weld could be proposed based on experimental measurements... The material model is chosen without deeper discussion and without verification. The mesh is simple, regular. The contents are simulations of simple geometric shapes for the nonlinear material model. The simulation tool used - Abaqus - is a commonly available software. The structural node model used in the paper is not difficult and requires only the usual engineering skills. There is nothing scientific or new in the article. The reason is that this is not a scientific article but a technical report on strength analyses.

In terms of terminology - principal plastic deformation is principal plastic strain. It would be interesting to know its direction as well. However, the stresses are only given by von Mises...

Response

The paper presented is very innovative and presents the results of a numerical study on a new joint component, which is not contemplated in the current version Eurocodes. The authors understand the observations of the reviewers, nevertheless the paper must be assessed under a different point of view, which is that of people doing research in the field of steel structures and, more specifically the modelling of joints with Eurocode 3 part 1.8. The research is strongly applicative and provides a formula to be included in this code.

The novelty regards the modelling of new typologies of joints, realized with a new technology for the cutting of steel materials, which the 3D laser cutting. This topic belongs to the application of traditional construction materials in new forms and is consequently in line with the purpose of the special issue for which it has been submitted (Conventional vs. Innovative Materials, Tradition and Innovation). Currently, these components have not been studied yet, and no formulations are included in Eurocodes. Consequently, the paper's originality is the definition of the strength and stiffness of the investigated components and their validation against numerical outcomes deriving from a parametric analysis. Regarding the FE model, it has been developed relying on a strong background of the modelling of similar components. Additional information have been added in the paper and we hope that the reviewer can frame better the activity done. As a guest editor of this special issue, I believe that the topic treated fits perfectly. I am not pretty sure about the judgment given and I request a review about the technical content, rather than a subjective opinion about the topic.

Reviewer 3 Report

The English of the text is good with a few minor grammar errors and typos. A check is advised.
The manuscript is logically organized and in most parts easy to follow.
Regarding the methodology, there are some issues that should be addressed:
1. The input action is the rotation of the beam, as shown clearly in Fig.3. This results in parts of the flange to press longitudinally onto the column wall (red, buckling, Ftc), and other parts detach from it (blue, tearing, Ftt). In the numerical simulation, the authors considered only a translation of the flange perpendicular to its plane and longitudinal with respect to the column (Fig.8). No rotation is considered, though. I would regard it more realistic if the applied actions include both. The authurs should explain why the rotation can be omitted or prove by a comparative finite element simulation in a typical case. It may turn out that rotation has no significant effect on the results, though it would be better if it was justified.
2. Regarding the formula fitting, the authors have chosen beta^c1*gamma^c2. The results have coefficient of variation between 6.6% and 14.4%. It not unacceptably large, but one may ask if any better fit could be achieved with other types of formulas instead of the power expression. Consideration of other options may be useful.

Overall assessment:
After appropriate explanations/amendments, the manuscript can be considered for publication.

Author Response

Comment 1

The input action is the rotation of the beam, as shown clearly in Fig.3. This results in parts of the flange to press longitudinally onto the column wall (red, buckling, Ftc), and other parts detach from it (blue, tearing, Ftt). In the numerical simulation, the authors considered only a translation of the flange perpendicular to its plane and longitudinal with respect to the column (Fig.8). No rotation is considered, though. I would regard it more realistic if the applied actions include both. The authurs should explain why the rotation can be omitted or prove by a comparative finite element simulation in a typical case. It may turn out that rotation has no significant effect on the results, though it would be better if it was justified.

Response 1

The following sentences have been added to the manuscript:

It is worth highlighting that the choice of applying displacements along the longitudinal direction only, without also the rotation of the plate is ascribed to the mechanical behaviour observed during the experimental campaigns carried out in [11]. In fact, as it has also been confirmed by numerical simulations (Figure 7), the part of the passing-through plate in the tubular profile is connected to the other nodal components and acts as a rigid plate; consequently, the actions transmitted by the beam become loads acting orthogonally to the plate. Moreover, the rotation of the flanges in the column is lower than the rotation exhibited at the beam-to-column attachment, justifying this effect can be ignored.

Comment 2

Regarding the formula fitting, the authors have chosen bc1g c2. The results have coefficient of variation between 6.6% and 14.4%. It not unacceptably large, but one may ask if any better fit could be achieved with other types of formulas instead of the power expression. Consideration of other options may be useful.

Response 2

The chosen fitting formulations have been defined considering that only b and g are the parameters affecting the response of the analysed components. Figures 11 and 12 show that exponential laws represent the best solution to define the mechanical modelling of the components. Moreover, it is necessary to provide very simple formulations to make them easily applicable in common practice.

Finally, it has been observed in the previous works carried out by the same authors that the strength and stiffness formulations to predict the mechanical behaviour of the components pcc/pct are expressed by exponential laws derived from analytical formulations. Consequently, to be consistent with this evidence and considering that the proposed formulations do not rely on analytical theories, exponential laws have also been proposed for the analysed cases (ttt/ttc). 

Round 2

Reviewer 1 Report

Congratulations,

Figures 2 and 3 presented in current version of the manuscript have enough design changes to be significantly different compared to the ones published in yours previous studies.

All requested corrections and completions were well effectuated. The current version of the manuscript is properly improved.

Author Response

The authors intend to express their gratitute to the reviewer for the help provided in improving the quality of the paper

Reviewer 2 Report

The article has improved in title only on the recommendation of the second reviewer.
The level of simulations does not represent a scientific approach to the problem. The authors consider ideal geometry and tabular material properties. They mention 3% imperfection, but it is not clear to which geometric dimension it is applied or how much effect it has on the resulting stresses. The Young's modulus of elasticity in tension for this type of low carbon steel can have a value from 180000 - 200000 MPa and combined with various deviations from the ideal geometry (wall thickness, dimensions and flatness of profiles, accuracy of hole cutting, ...) they can produce many possible results that should be validated by experiment and based on it a procedure for modelling such joints would be recommended and reference curves for strength and stiffness would be established. Also, the analysis of the use of other criteria for equivalent stresses (e.g. Tresca's criterion) including the change of Poisson's number in the plastic region is missing.  The paper remained a technical study without a scientific approach and is not suitable for a journal with a focus on materials research.

Author Response

The present paper deals with the study of the mechanical behaviour of two components belonging to beam-to-column connections between Circular-Hollow-Section columns and passing-through double-tee beams. The methodology employed to achieve this aim consists in:

i) identifying the main components of the joint;

ii) performing experimental tests on the components and beam-to-column sub-assemblies;

iii) modelling the tested specimens through one or more Finite Element software and validating them against the experimental results;

iv) exploiting the validated FE models to perform parametric analyses on more comprehensive sets of sub-assemblies;

v) assessing the accuracy of the proposed formulations against the results of the parametric analyses;

vi) proposing theoretical formulations to characterise the nodal components’ strength and stiffness and combine them to obtain the flexural response of the whole joint.

The main sources of deformability of the joint have been identified in [22], and some of them (specifically pcc/pct) have been analysed from an experimental, numerical and analytical point of view. In particular, tests have been carried out on CHS to axially loaded passing-through plates, and FE models have been validated against the experimental results.

In order to complete the mechanical characterisation of the nodal components, this paper investigates the behaviour of the sources of deformability defined as ttt/ttc exploiting the numerical models of CHS to plate connections validated in the previous phase and modifying the way in which the displacement histories are applied.

Based on this assumption, a parametric analysis of numerically simulated configurations has been carried out and has allowed discovering relationships among the connections’ material and geometrical parameters and their mechanical behaviour.

This explanation demonstrates that the present work represents a part of a more exhaustive study based on a robust scientific approach.

The reviewer comments that ideal geometry and tabular material properties are employed. The authors have performed the analyses starting from realistic configurations of connections representative of the flanges of CHS to passing-through double-tee beam joints, according to a similar approach applied in previous studies [11, 12, 22]. Consequently, the parametric analysis is full of cases that designers could exploit in practice. Furthermore, since the aim is to provide design formulations, it is obvious that the derived equations and the simulations are performed using tabular material properties because these are the values used in practice by designers. It is worth highlighting that the material properties are clearly discernible in the proposed formulations and can be appropriately modified by designers if they adopt configurations characterised by different materials. The proposed formulations can be applied if different geometries and materials are selected, proving the present work has general applicability.

It is clearly written in the paper that: “the buckling mode is amplified according to the requirements of Eurocode 3 part 1.5 [27] and EN 10034 [28] with a scale factor reproducing an imperfection on the tube equal to the 3% of the nominal internal diameter (Figure 6)”. This parameter is important in order to introduce in the model the geometric non-linearities that, otherwise, could be neglected. The same approach has been applied for the components pcc/pct and confirms the scientific approach to the problem.

The Young’s modulus mainly affects the elastic response of the component and is assumed equal to 210 GPa, representing the nominal value commonly used in research and practice and suggested by Eurocode 3 part 1-1 (see this figure, extracted from the Eurocode 3 part 1-1).

There is no need to study the variation of this parameter because it is explicitly reported in the proposed formulation.

Considering the previous statements, the authors believe that the present paper provides additional information which can be helpful for the exploitation of CHS with passing-through elements joints in practice. Consequently, this work represents an application of traditional construction materials in new forms.

Based on these considerations, the authors kindly ask the reviewer to reconsider the opinion related to the proposed manuscript.
